# Synthesis, Transfer, and Gas Separation Characteristics of MOF-Templated Polymer Membranes

**DOI:** 10.3390/membranes9100124

**Published:** 2019-09-20

**Authors:** Sophia Schmitt, Sergey Shishatskiy, Peter Krolla, Qi An, Salma Begum, Alexander Welle, Tawheed Hashem, Sylvain Grosjean, Volker Abetz, Stefan Bräse, Christof Wöll, Manuel Tsotsalas

**Affiliations:** 1Institute of Functional Interfaces (IFG), Karlsruhe Institute of Technology (KIT), Hermann-von Helmholtz-Platz 1, 76344 Eggenstein-Leopoldshafen, Germany; Sophia.Schmitt@web.de (S.S.); peter.krolla@kit.edu (P.K.); qi.an@kit.edu (Q.A.); alexander.welle@kit.edu (A.W.); tawheed.hashem@kit.edu (T.H.); christof.woell@kit.edu (C.W.); 2Institute of Polymer Research, Helmholtz-Zentrum Geesthacht (HZG), Max-Planck-Street 1, 21502 Geesthacht, Germany; sergey.shishatskiy@hzg.de (S.S.); volker.abetz@hzg.de (V.A.); 3Karlsruhe Nano Micro Facility (KNMF), Karlsruhe Institute of Technology (KIT), Hermann-von-Helmholtz-Platz 1, 76344 Eggenstein-Leopoldshafen, Germany; 4Institute of Physics and Technology, International X-ray Optics Lab, National Research Tomsk Polytechnic University (TPU), 30 Lenin ave., Tomsk 634050, Russia; 5Institute for Organic Chemistry (IOC), Karlsruhe Institute of Technology (KIT), Fritz-Haber-Weg 6, 76131 Karlsruhe, Germany; sylvain.grosjean@kit.edu (S.G.); stefan.braese@kit.edu (S.B.); 6Soft Matter Synthesis Lab, Institute of Biological Interfaces 3 (IBG3), Karlsruhe Institute of Technology (KIT), Hermann-von Helmholtz-Platz 1, 76344 Eggenstein-Leopoldshafen, Germany; 7Institute of Physical Chemistry, University of Hamburg, Martin-Luther-King-Platz 6, 20146 Hamburg, Germany; 8Institute of Toxicology and Genetics (ITG), Karlsruhe Institute of Technology (KIT), Hermann-von Helmholtz-Platz 1, 76344 Eggenstein-Leopoldshafen, Germany

**Keywords:** thin films, gas separation, selective layer transfer, metal–organic framework (MOF)

## Abstract

This paper discusses the potential of polymer networks, templated by crystalline metal–organic framework (MOF), as novel selective layer material in thin film composite membranes. The ability to create mechanically stable membranes with an ultra-thin selective layer of advanced polymer materials is highly desirable in membrane technology. Here, we describe a novel polymeric membrane, which is synthesized via the conversion of a surface anchored metal–organic framework (SURMOF) into a surface anchored gel (SURGEL). The SURGEL membranes combine the high variability in the building blocks and the possibility to control the network topology and membrane thickness of the SURMOF synthesis with high mechanical and chemical stability of polymers. Next to the material design, the transfer of membranes to suitable supports is also usually a challenging task, due to the fragile nature of the ultra-thin films. To overcome this issue, we utilized a porous support on top of the membrane, which is mechanically stable enough to allow for the easy membrane transfer from the synthesis substrate to the final membrane support. To demonstrate the potential for gas separation of the synthesized SURGEL membranes, as well as the suitability of the transfer method, we determined the permeance for eight gases with different kinetic diameters.

## 1. Introduction

Membranes are the key tools for numerous industrial separation processes [1,2,3,4]. Up to now, a vast number of synthetic membranes have been applied for separation and purification of liquids and gases [2]. The success of a membrane application for e.g., gas separation, depends on two main parameters, high permeance of the membrane for a specific component and high selectivity toward the desired compound from a mixture of compounds. While the selectivity of the membrane is controlled mostly by the material of the selective layer, the achievable membrane permeance on the other hand depends significantly on both the material property as well as the physical parameters of the selective layer, i.e., the layer thickness.

Microporous membranes [5] with pore sizes less than 2 nm showed good results as molecular sieving materials for gas storage and separation [6,7,8]. Inorganic or hybrid microporous materials, such as metal–organic frameworks (MOFs) or zeolites, feature excellent potential for molecular sieving applications due to their controllable pore architecture [9,10]. One major obstacle for the actual use of these materials for membrane applications is the difficulty to generate large area, defect free membranes with a thin selective layer, due to the materials’ crystalline and brittle nature. The fabrication of mixed matrix membranes (MMM), where MOFs [11,12,13], COFs [14], or other porous materials are embedded in a polymer matrix is one approach to overcome this problem, and shows promising results in gas separation, but does not overcome the issue of the selective layer thickness reduction [15,16,17,18,19]. To address this problem, we used a modular approach, where we first synthesized a crystalline SURMOF thin films [20] via a layer-by-layer (LbL) synthesis approach, which we then converted into a polymer network (SURGEL) nanomembrane [9,21], while keeping the molecular structure of the MOF template (see Figure 1).

The LbL synthesis approach allows fabrication of homogeneous SURGEL membranes with nanometer-sized layer thicknesses. In addition, this approach allows to further functionalize the internal and external surface of the membranes [22,23,24]. The highly variable chemical composition of SURGELs and their high thermal stability and excellent chemical inertness make them promising candidates for applications in gas separation, nanofiltration, and membrane catalysis [25,26]. Critical features to be considered in the synthesis and processing of ultra-thin membranes are their mechanical and chemical stability and the possibility to form continuous films with minimal defect rates [27]. Defects can be generated during the synthesis or during the transfer procedure (e.g., rupture, overlapping, or folding).

For gas transport experiments, we prepared SURGEL thin film composite (TFC) membranes and compared two different transfer methods, using either a continuous or a porous coating layer to provide the necessary mechanical stability during the transfer of our SURGEL layer from the synthesis substrate to the highly permeable polymeric substrate. While the continuous transfer layer needs to be removed after the transfer, the porous transfer layer—with a high surface porosity—can remain on the membrane as a protective layer. [28] We determined the permeances of our SURGEL TFC membranes for eight gases with different kinetic diameters. This paper demonstrates the potential of SURGELs for gas separation.

## 2. Materials and Methods

Chemicals: All chemicals were purchased from commercial sources and were used without any further purification, if not indicated differently. Copper (II) acetate monohydrate, 11-mercapto-1-undecanol (MUD) and toluene were purchased from Sigma Aldrich, Darmstadt, Germany. Layer-by-layer synthesis was performed in ethanol (absolute for analysis EMSURE), was purchased from VWR, Bruchsal, Germany. The 2,2′-diazido-4,4′-stilbenedicarboxylic acid (DA-SBDC) linker and the multifunctional alkyne cross-linker were synthesized as described elsewhere [9].

Substrates: Gold on mica sheets (Georg-Albert-PVD, Karlsruhe, Germany) were used as substrates for MOF synthesis. The SURGEL thin films were transferred onto a supporting membrane developed by Helmholtz-Zentrum Geesthacht and consisting of a thin layer of polydimethylsiloxane (PDMS) on a polyacrylonitrile (PAN) ultrafiltration membrane, supported by a non-woven polyester for gas permeation experiments, as described elsewhere [22].

Self-assembled monolayer (SAM): A clean gold-coated Mica substrate was immersed in a 20 µM solution of MUD in ethanol for 24 h. Afterwards, the substrate was rinsed thoroughly with ethanol and dried in a nitrogen stream.

Preparation of SURGEL films: The SURGEL films were prepared as described previously [9,22]. The following steps were involved: SURMOF synthesis, SURMOF cross-linking, and metal removal to convert the SURMOF into a SURGEL.

Transfer of the SURGEL membrane using continuous PMMA: To transfer the SURGEL membranes, first a continuous layer of PMMA was spin coated on top of the SURGEL as a supporting layer. Afterwards, the gold layer between Mica and SURGEL was etched away to detach the Mica from the SURGEL/PMMA film. After washing, the membrane was transferred to the desired substrate and the PMMA was dissolved by immersion in acetone. High molecular weight of PMMA allows one to consider the absence of polymer penetration into pores of the SURGEL. (For a detailed description see Lindemann et al. [29].)

Transfer of the SURGEL membrane using porous PMMA: To transfer the SURGEL membranes, first a mixture of PMMA/PS = 8/2 *wt*/*wt* in dichloromethane was spin coated on top of the SURGEL. Afterwards, the PS regions were selectively dissolved by immersion into cyclohexane for 24 h. Etching of the gold layer between Mica and SURGEL enabled detachment of the Mica from the SURGEL/por-PMMA film. After washing, the membrane was transferred to the desired substrate.

Infrared Reflection Absorption Spectroscopy: The infrared reflection absorption spectroscopy (IRRAS) spectra were recorded using a VERTEX 80 (Bruker GmbH, Karlsruhe, Germany) purged with dried air. The IRRAS accessory (A518) has a fixed angle of incidence of 80°. The data were collected on a narrow-band liquid-nitrogen-cooled mercury cadmium telluride detector. Perdeuterated hexadecanethiol-self-assembled membranes (SAMs) on Au/Ti/Si were used for reference measurements.

X-ray Diffraction: The X-ray diffraction (XRD) data were measured using a diffractometer (D8-Advance Bruker AXS, Karlsruhe, Germany) with Cu Kα radiation (λ = 1.5418 Å) in θ–θ geometry and with a position-sensitive detector and variable divergence slit.

Scanning Electron Microscopy (SEM): Prior to SEM measurement, an approximately 5-nm-thick gold–platinum layer (Bal-Tec/MED020 Coating System, Macclesfield, United Kingdom) was coated on the samples to improve their conductivity. The prepared samples were imaged with an environmental scanning electron microscope equipped with an SE detector using an acceleration voltage of 10–20 keV (FEI Philips XL30 FEG-ESEM, Philips, Amsterdam, The Netherlands).

Atomic force microscopy (AFM): An Asylum Research Atomic Force Microscope, MFP-3D BIO (Asylum Research, High Wycombe, United Kingdom), was used for imaging. The AFM was operated in an isolated chamber at 25 °C in alternating current (AC) mode. AFM Ultrasharp™ cantilevers from MikroMasch (provided by NanoAndMore, Wetzler, Germany) were used.

Time-of-flight secondary ion mass spectrometry (ToF-SIMS)*:* Measurements were performed on a TOF.SIMS5 instrument (ION-TOF GmbH, Münster, Germany). This spectrometer was equipped with a bismuth cluster primary ion source and a reflectron type time-of-flight analyzer. UHV base pressure was <5 × 10^−9^ mbar. For high mass resolution, the Bi source was operated in the “high current bunched” mode providing short Bi_3_^+^ primary ion pulses at 25 keV energy and a lateral resolution of approximately 4 μm. The short pulse length of 1.1 to 1.3 ns allowed for high mass resolution. Primary ion doses were kept at 10^11^ ions/cm^2^ (static SIMS limit) for these measurements. Due to the highly insulating nature of the sample, charge compensation during spectrometry was necessary. Therefore, an electron flood gun providing electrons of 21 eV was applied, and the secondary ion reflectron tuned accordingly. Spectra were calibrated on the omnipresent C^−^, CH^−^, CH_2_^−^, C_2_^−^, C_3_^−^, or on the C^+^, CH^+^, CH_2_^+^, and CH_3_^+^ peaks. Based on these datasets the chemical assignments for characteristic fragments were determined. For measurements with high lateral resolution, the primary ion gun was operated in “burst alignment” mode, providing long pulses (nominal mass resolution).

Gas transport measurements: All membranes were placed into a membrane holder equipped with an ethylene-propylene-diene-monomer rubber (EPDM, German: Ethylen Propylen Dien Monomer Kautschuk) sealing O-ring. EPDM was chosen as a material of the sealing O-ring from a number of other materials as one not allowing any low molecular compound (e.g., plasticizer) to leach out, and thus not contaminating the few nanometer thick selective layer of the membrane, which is highly important for materials working as molecular sieves such as carbon nanomembranes (CNM) or conjugated microporous polymers (CMP) nanomembranes [22,30]. In order to ensure that the EPDM O-rings were free of any low molecular weight compound, they were washed with acetone in a Soxhlet extractor. For all membrane samples, the permeances of H_2_, He, CO_2_, O_2_, N_2_, Ar, CH_4_, and C_2_H_6_ were measured at feed pressures of 110 to 280 mbar and in the temperature range 70–30 °C using a constant volume/variable pressure experimental setup, as described in detail elsewhere [31,32]. Sufficiently low feed pressures were chosen to employ the ideal gas assumption for the data evaluation, while being high enough to match the measurement system sensitivity. The result was considered acceptable if the permeance values scattered less than 5% in ten repeated measurements for each gas.

## 3. Results and Discussion

### 3.1. Preparation of Freestanding SURGEL Membranes

For the application of SURGEL material in membrane separation, a crucial step is the preparation of a nanometer thick membrane free of large defects, originating from synthesis imperfections or membrane integration procedure within thin film composite (TFC) membranes. Synthesis of SURGEL membranes is based on layer-by-layer synthesized azide-functionalized metal–organic framework on functionalized gold-coated Mica substrates. The azide groups in the SURMOF structure enable the crosslinking with a multitopic secondary linker. Subsequently, the copper ions of the, now fully cross-linked, scaffold can be removed with ethylenediaminetetraacetic acid (EDTA) to convert the SURMOF into a polymeric few-nanometer-thick film (SURGEL) [9]. XRD and IRRAS characterization of the different steps are shown in Appendix A. To prepare freestanding SURGEL films, we employed sacrificial gold-coated Mica substrates, which allowed to transfer the SURGEL films onto a PDMS/PAN support for gas permeation measurements [30,33]. For the gas permeation measurements, we pursued two different strategies regarding the transfer approaches. In the first approach, we coated the SURGEL with a continuous layer of pure PMMA to protect the fragile membrane during the transfer procedure. After placing the SURGEL/PMMA film on the PDMS/PAN support, we dissolved the PMMA film in acetone to receive the bare SURGEL membrane on the support. This strategy worked successfully for the PDMS-coated PAN support, since the SURGEL was directly placed onto the homogenous PDMS surface and securely fixed on it by van-der-Waals force, preventing the detachment during the acetone treatment to dissolve the PMMA transfer layer. However, the acetone treatment to dissolve the PMMA transfer support also imposes limitations towards possible support substrates (stability issue towards acetone) and limits the upscaling possibilities. Therefore, we followed a second strategy, where we used a porous PMMA (por-PMMA) transferring method [34]. To prepare the por-PMMA layer, we spin coated a mixture of poly(methyl methacrylate)/polystyrene (PMMA/PS) in dichloromethane onto the SURGEL film. The evaporation of solvent during spin coating of polymer mixtures leads to a phase separation of the polymer blend, if two immiscible polymers are used. Afterwards, we placed the obtained composite material in cyclohexane for 24 h to completely dissolve the PS domains, leaving a mechanically-stable highly-porous PMMA transfer support on top of the SURGEL film. The por-PMMA support allows a stable transfer of the SURGEL film without risking that organic solvents would alter the state of the PDMS layer by e.g., swelling. Figure 2 shows SEM images of freestanding SURGEL membranes, supported by por-PMMA.

The 2–4 µm round-shaped pores of the PMMA film, which are completely covered by the continuous SURGEL membrane, are clearly visible in the SEM images and no large defects in the SURGEL membrane are visible. From the cross section image in Figure 2c, we estimated the thickness of the por-PMMA support to about 500 nm, while the SURGEL membrane is an order of magnitude thinner (about 50 nm).

To evaluate the thickness of the transferred SURGEL membranes more accurately, we used atomic force microscopy (AFM) for taking images at the edge regions of the membrane. Figure 3 shows two representative AFM images of the SURGEL/por-PMMA membranes. Across the edges of the SURGEL/por-PMMA membranes, we performed line scans (indicated by the red lines in the images at the left side). On the right of the two images are the corresponding height profiles along the red lines. The thickness of the SURGEL layers are given as dZ values above the height profiles.

The AFM images and height profiles suggest a thickness of the SURGEL layer of roughly 50 nm (±5 nm) and a thickness of the por-PMMA layer of about 500 nm (±50 nm), which matches very well with the thickness estimation from the SEM images of Figure 2.

The presence of PMMA as the topmost sample layer is evidenced by ToF-SIMS. All fragments characteristic for this polymer, especially CH_3_O¯ and C_4_H_5_O_2_¯, are detectable. On the millimeter scale, the por-PMMA layer is fairly homogeneous without larger defects, see Figure 4a,b. Imaging of the pores in this layer, shown in Figure 4c, is performed by applying a non-bunched imaging SIMS mode to avoid chromatic aberration of the primary beam. Further, it was ensured that signals of CH_3_O¯ and C_4_H_5_O_2_¯, characteristic for PMMA, and CN^−^, characteristic for the underlying SURGEL layer, were not accompanied by strong side peaks on the same nominal masses.

Scanning a 50 × 50 µm^2^ field of view and recording 512 × 512 pixel, high lateral resolution images were obtained and combined into the red/green overlay (Figure 4c), showing the chemical contrast of SURGEL derived fragment CN^−^, green, and PMMA-derived fragments, red. Since the information depth of SIMS is in the range of a few nanometers, we concluded that the PS phase was fully removed during the solvent treatment and the pores in the topmost por-PMMA layer were open.

The possibility to transfer the SURGEL membrane using a highly-porous, but still stable, transfer support layer of PMMA allowed us to directly use the por-PMMA/SURGEL composite system for the gas separation, without the need of an additional dissolution step in acetone, thus avoiding possible swelling of the PDMS layer, followed by shrinking when acetone evaporates. For the gas separation experiments, we employed both transfer techniques (por-PMMA and continuous PMMA) to compare the gas separation properties of the two transfer methods.

### 3.2. Gas Permeation Experiments

For gas permeation experiments, we prepared thin film composite (TFC) membranes [28,35], comprised of PDMS/PAN and SURGEL, as selective layer. We chose the PDMS/PAN support to enhance the mechanical stability to the nanometer-thick SURGEL and minimize roughness-induced strain [30]. The PDMS layer also diminishes the effect of viscous gas flow through larger defects in the SURGEL layer. To evaluate the membrane properties of the individual SURGEL layer, we utilized the resistance model introduced by Henis and Tripodi [36]. In previous works, such an analysis has been successfully applied for the determination of properties of nanometer-thin CNMs [30] and CMPs [22]. We determined the gas transport properties of the reference and TCF-SURGEL samples (PDMS/PAN, SURGEL/PDMS/PAN and por-PMMA/SURGEL/PDMS/PAN) for He, H_2_, CO_2_, Ar, O_2_, N_2_, CH_4_, and C_2_H_6_. We chose these gases to cover a big range of kinetic diameters and molecular weights. Higher hydrocarbons were not used to avoid possible swelling of PDMS or the SURGEL layer during the exposure to the gas, which can damage the membranes. The measured gas permeances are presented as a function of penetrant kinetic diameters in Figure 5.

For these measurements, two SURGEL membranes of the same batch were transferred by the different procedures described above, to create the SURGEL/PDMS/PAN TFC membranes. The gas permeances of SURGEL/PDMS/PAN membranes were significantly lower compared to the reference PDMS/PAN membranes alone, indicating that the SURGEL layer on top of the PDMS dominated the gas transport through the membrane (please note the different scales for SURGEL (right) and reference samples (left) to fit into one figure).

The porosity of the por-PMMA was evaluated by comparing the gas transport properties of por-PMMA/PDMS/PAN and PDMS/PAN membranes. As it can be seen from Figure 5, the permeance profiles for bare PDMS/PAN and PDMS/PAN covered with porous PMMA (por-PMMA) layer are mostly similar. It indicates that due to the rigidity of the por-PMMA layer, the contact to the PDMS is most probably not perfect due to the roughness of the PDMS surface and por-PMMA rigidity, allowing the gas to be transported laterally under the PMMA layer, which consequently leads to an overestimation of the por-PMMA layer porosity. A much stronger effect of por-PMMA on the transmembrane transport can be observed for the case of SURGEL membranes. Considering only data for helium permeance, as smallest penetrant of the study, one can estimate the surface porosity of the por-PMMA layer as 33% and for all gases as 30 ± 6%, while por-PMMA/SURGEL the corresponding values are 79% and 77 ± 7%. Based on these results, we decided to use the PDMS/PAN membrane as reference to evaluate the properties of both SURGEL/PDMS/PAN and por-PMMA/SURGEL membranes. For por-PMMA/SURGEL, the gas transport properties were reduced to match 33% of membrane area available for gas transport. The information on gas transport properties of the defect free SURGEL layer was extracted from the experimental results obtained for the SURGEL/PDMS/PAN TFC membrane using the Henis and Tripodi resistance model [37], where different layers of the membrane are considered as resistors in series.

The values obtained by the permeance measurement (*P*_SURGEL/PDMS/PAN_) can be considered as combined values for the stacked system SURGEL-membrane (*P*_SURGEL_) and PDMS/PAN-membrane (*P*_PDMS/PAN_). The relationship between the permeances of the combined layers to the permeances of the individual layers is:1/*P*_SURGEL/PDMS/PAN_ = 1/*P*_SURGEL_ + 1/*P*_PDMS/PAN_

Figure 6a shows the permeance of the TFC composite membrane and the individual SURGEL layer alone, indicated with (RM) for the resistance model. Figure 6b shows the selectivities of the TFC membranes and the individual SURGEL layers after applying the series resistance model.

Several conclusions can be drawn from the data presented in Figure 6a. First, the application of the resistance model yields the same results for the He and H_2_ permeances, allowing us to draw the following conclusions: (I) The properties of these two gases can be used for estimation of the por-PMMA porosity; (II) the pores of the SURGEL membrane are large enough to allow diffusion of He and H_2_; (III) SURGEL and por-PMMA/SURGEL show clear molecular sieving for CO_2_ and other bigger molecules—for por-PMMA/SURGEL, efficient sieving was observed for CO_2_, in fact the carbon dioxide permeance became quite comparable to other larger penetrants; and (IV) the descending trend of the por-SURGEL permeance for gases larger than O_2_ indicates a lower number of defects (app. 1.8%) compared to SURGEL membrane (2.3%). This low defect number drastically reduces the influence of the PDMS support on the data obtained after application of the resistance model. For example, there is an ascending trend between CH_4_ and C_2_H_6_ on the SURGEL dependence, and a descending trend on the por-PMMA/SURGEL dependence.

The selectivities for hydrogen over other gases shown in Figure 6b are similar for the two types of SURGEL membranes. The ideal selectivities for H_2_/N_2_ of 16.0 for SURGEL/PDMS/PAN and 13.5 for por-PMMA/SURGEL/PDMS/PAN are in a similar range as reported for e.g., ZIF-8 membranes 11.6 (H_2_/N_2_) [38] or ZIF-70 of 15.8 (H_2_/N_2_) [39]. It is interesting to note that SURGEL outperforms por-PMMA/SURGEL for gases consisting of atoms or molecules larger than Ar. The reason may be related to the small molecular sieving effect of the por-PMMA/SURGEL between He and H_2_, which is less significant for the unsupported SURGEL. In summary, both SURGEL and por-PMMA/SURGEL membranes show characteristics typical for glassy polymer membranes with stiff polymer backbones [40,41,42,43].

Considering the roughly 50 nm thickness of the SURGEL layer, one can estimate the permeability coefficient of H_2_ as ~3.7 Barrer, which is moderate, not among the most permeable polymeric materials studied to date. Taking into account the high variability in the SURGEL synthesis and the procedure of layer growth, as well as the optimized transfer of nanometer-thick layer from the synthesis substrate to gas transport supports, achieving a membrane permeance in the range above 1 Nm^3^m^−2^ h^−1^ bar^−1^ appears to be a realistic goal.

### 3.3. Activation Energies of the SURGEL Membranes

The measurements of gas transport properties carried out in the temperature range 70–30 °C allowed us to evaluate activation energies of permeance for all studied gases (measured gas permeances at each temperature are listed in the Appendix A). It was found that the application of the resistance model to the experimental data does not disturb the linear behavior of the Arrhenius plot, and from the slope of the dependence of ln(L) vs. T^−1^ (K^−1^) the activation energy can be acquired with sufficient accuracy. The obtained values of activation energies of permeance for different gases are plotted against the kinetic diameter of the penetrant molecules in Figure 7.

The dependences of the permeance activation energy on the kinetic diameter of the tested gases for the SURGEL and por-PMMA SURGEL membranes has a linear character and ascends with increasing penetrant molecule diameter. A similar trend was demonstrated by Stannet et al. [44] for the activation energy of diffusion vs. molecular diameters of penetrant in glassy polyvinyl acetate. The linear correlation curve depicted in Figure 7 is very similar for both SURGEL and por-PMMA/SURGEL. Activation energies for H_2_ and He are very similar for two membranes, but there is significant difference in the Ea for CO_2_, the value for the SURGEL is significantly lower than that of por-PMMA SURGEL, but at the same level as H_2_ and He. The low Ea for these three gases correlates well with the gas permeance data (Figure 6), where both SURGEL membranes show significant drops of gas permeances for gases larger than CO_2_. Therefore, we can assume that the resistance model has its limitations in extraction of the properties of the selective layer of interest from the properties of the whole membrane. It was demonstrated earlier that a clear correlation exists between the fractional free volume (FFV) of the glassy polymer and activation energies of both permeability and diffusion: Higher FFV leads to lower activation energy [45]. The values of activation energies obtained for the SURGEL membranes underline, again, the conclusion that the SURGEL membranes show highly molecular size-dependent gas transport properties.

## 4. Conclusions

In conclusion, our results demonstrate that highly-stable freestanding SURGEL membranes can be successfully fabricated via layer-by-layer synthesis of SURMOFs, followed by crosslinking of the organic linkers. This strategy to fabricate SURGEL membranes enables the creation of laterally-structured membranes with tailorable properties via chemical functionalization. The gas permeation experiments of the prepared SURGEL membranes show highly molecular size-dependent gas transport properties, in clear contrast to the support membranes. The successful preparation of SURGEL thin films and freestanding membranes, combined with the ability of a post-synthetic functionalization, can be developed into a kind of “tool box” to create tailored SURGEL membranes. This study paves the way for the creation of more complex SURGEL membranes, including vertically-organized pore geometries, as well as incorporation of functional groups to tune the permeance and selectivity for applications in liquid and gas separation.

In addition, we used a new transfer method for nanomembranes using a highly porous transfer support. With this new transfer technique, the nanomembranes can be used directly after transfer, avoiding the usually necessary dissolution step. This new method allows an easier upscaling in the integration of nanomembranes into gas and liquid phase separation processes, and avoids possible membrane damage associated with the transfer support dissolution step.

The new membrane material and transfer method show great promise to be integrated in large-scale gas separation devices. The described methods can also allow the integration of nanomembranes into other areas, such as sensing, catalysis, or optical/electronic applications, since the membranes can be transferred to various other substrates. Also, the possibility to remotely control the selectivities by incorporating photo-switchable molecules [46] into the SURGELs is possible.

## Figures and Tables

**Figure 1 membranes-09-00124-f001:**
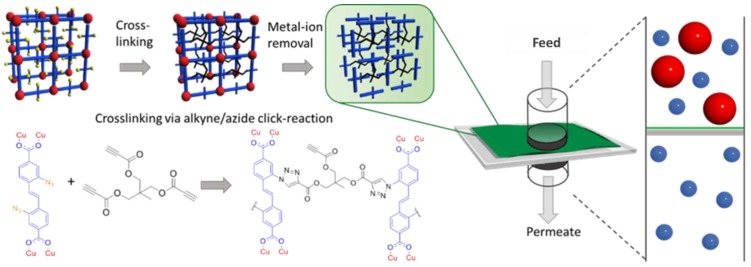
Synthesis of the SURGEL via crosslinking the azide functionalities within the SURMOF structure with a multitopic alkyne cross-linker.

**Figure 2 membranes-09-00124-f002:**
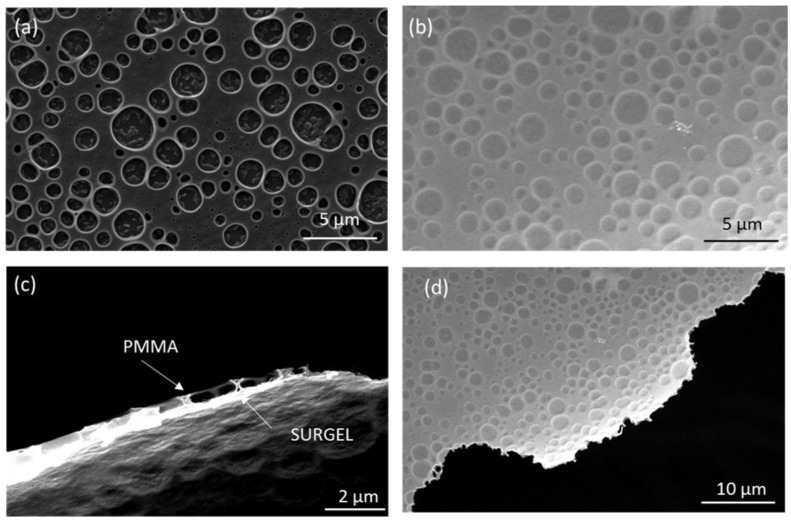
Scanning electron microscopy (SEM) images of SURGEL membranes supported by a por-PMMA layer: (**a**) Top view, (**b**) bottom view, (**c**) cross section at an edge region, and (**d**) bottom view at an edge region.

**Figure 3 membranes-09-00124-f003:**
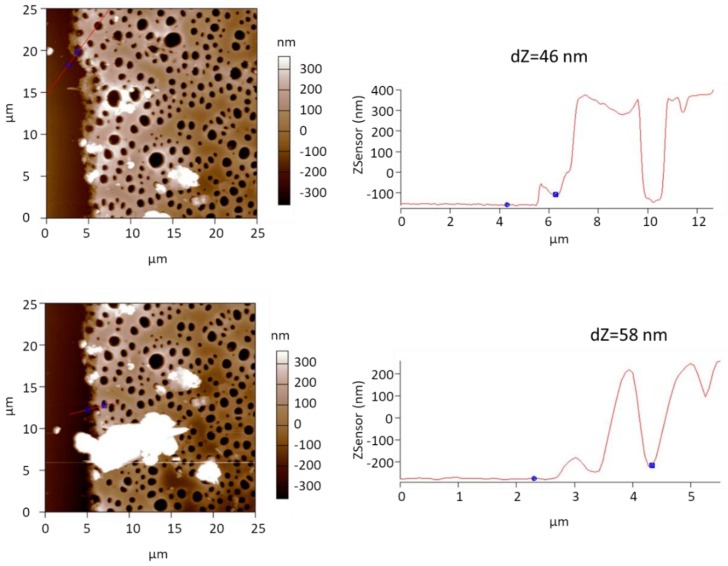
**Left**: Two representative atomic force microscopy (AFM) images of the SURGEL/por-PMMA membranes on top of the PAN/PDMS support. **Right**: Corresponding height profile along the red lines of the two images. The difference in height from the PDMS support to the SURGEL layer is given as dZ value above the height profiles.

**Figure 4 membranes-09-00124-f004:**
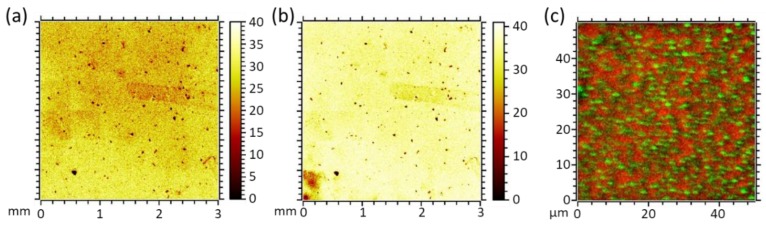
ToF-SIMS images of two characteristic PMMA fragments: (**a**) CH_3_O¯; (**b**) C_4_H_5_O_2_¯; (**c**) ToF-SIMS red/green overlay image. Green: CN^−^, 26 m/z. Red: Sum of CH_3_O¯, 31 m/z, and C_4_H_5_O_2_¯, 85 m/z.

**Figure 5 membranes-09-00124-f005:**
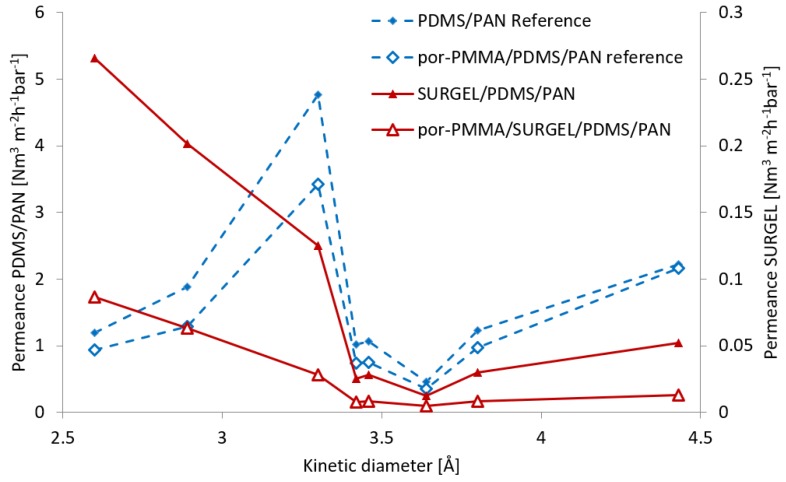
Measured gas permeances presented as a function of the penetrant kinetic diameters of the gases for the PAN/PDMS support membrane, the TFC membranes containing SURGEL and SURGEL + por-PMMA support on top, as well as the reference sample of por-PMMA on top of PAN/PDMS. Please note the different scales for SURGEL (**right**) and reference samples (**left**).

**Figure 6 membranes-09-00124-f006:**
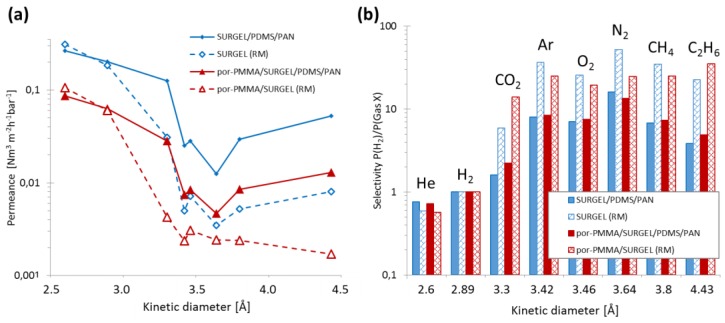
(**a**) The permeance of the TFC SURGEL membranes and the calculated permeances of the SURGEL layers alone, by applying the series resistance model (RM); (**b**) selectivities of the TFC SURGEL membranes and the individual SURGEL selective layers after applying the series resistance model (RM).

**Figure 7 membranes-09-00124-f007:**
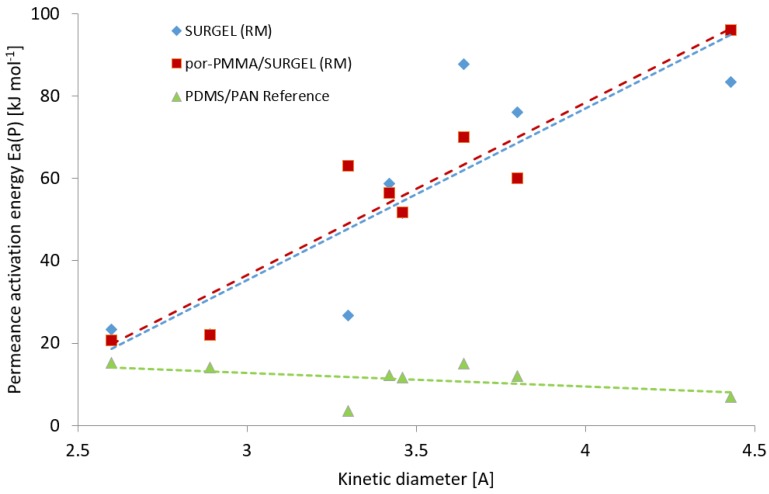
Dependencies of the permeance activation energy on the kinetic diameter of the tested gases for the PDMS/PAN membrane (**green triangles**), the SURGEL (RM) membrane (**blue diamonds**), and the por-PMMA/SURGEL (RM) membrane (**red squares**).

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
