# Peer review of "Synthesis, Transfer, and Gas Separation Characteristics of MOF-Templated Polymer Membranes"

_membranes, 2019, doi:10.3390/membranes9100124_

Round 1
Reviewer 1 Report
This MS describes an interesting fabrication method of gas separation membrane. The authors have used MOF as a template to construct polymer networks, and resultant membranes showed acceptable separation performance. However, there are some spots that lack clarity, and I believe that the following comments could help improve the quality of this study: 1. The SUREL was fabricated via crosslinking with a multitopic alkyne cross-linker, but there is no direct characterization to confirm the synthetic structure is the one shown in figure1. 2. The author has described that the SURGEL was fixed on the PDMS surface by van-der-Waals force in line 186. I think there should be a valid evidence to proof that such combination is reliable, because it will directly affect the results of later tests. 3. More characterizations about the MOF template should be provided to better explain the formation of SURGEL membranes, such as the EDS of membrane cross-section (SURMOF and SURGEL). 4. The references about hybrid membrane with microporous materials were not sufficiently cited and the authors can add some recent literatures like J. Membr. Sci., 570-571 (2019) 278-285;J. Mater. Chem. A, 7 (2019) 10898-10904 for enrich the content . 5. The coordinate axis of figures in this paper (especially fig.3 and fig.4) are very small and hard to recognize. 6. There is something wrong with the typesetting of paper, and please check the whole manuscript again and make some necessary improvements.
Author Response
We would like to thank the reviewers for their time to evaluate our manuscript for publication in Membranes. We highly appreciate all the reviewers’ valuable comments. In compliance with the reviewer’s comments and suggestions, we wish to address these points with additional experimental evidence where applicable and we have revised the manuscript accordingly.
Please find below a point-by-point answers and responses to the reviewer’s comments and suggestions; the reviewers’ comments are reproduced unchanged with our response included. We included a new supporting information filePoint-by-point response to reviewers:
Comments and Suggestions for Authors
This MS describes an interesting fabrication method of gas separation membrane. The authors have used MOF as a template to construct polymer networks, and resultant membranes showed acceptable separation performance. However, there are some spots that lack clarity, and I believe that the following comments could help improve the quality of this study:
The SUREL was fabricated via crosslinking with a multitopic alkyne cross-linker, but there is no direct characterization to confirm the synthetic structure is the one shown in figure1.
A1. We thank the reviewer for his comment and added XRD and IRRAS characterization of the SURMOF, the crosslinked SURMOF as well as the SURGEL to the supporting information Figures S1 and S2.
We added this information to the revised manuscript as follows:
“XRD and IRRAS characterization of the different steps are shown in supporting information Figures S1 and S2.”
Please note, that XRD measurements on the mica substrates was not possible due to strong peaks of the mica substrate itself. Therefore the XRD data was recorded on a reference substrate. However IRRAS and ToF-SIMS data are of the original sample on mica and on PAN/PDMS and confirm the synthesized membrane material.
The author has described that the SURGEL was fixed on the PDMS surface by van-der-Waals force in line 186. I think there should be a valid evidence to proof that such combination is reliable, because it will directly affect the results of later tests.
A2. The PDMS layer is fully cross-linked during the PDMS membrane formation and not able to react to any functional group of the SURGEL. The SURGEL deposition is performed in water meaning the PDMS layer cannot be affected by the solvent. E.g. swelling. It means the only force keeping the layers together is van der Waals as it was discussed in previous publications (see Chem. Mater. 2014, 26 (24), 7189-7193 and Adv. Mater. 2014, 26 (21), 3421-3426).
As correctly pointed out by the reviewer, we have to consider that the SURGEL layer is not perfect and can contain defects caused by the preparation or by deposition. To take such defects into account, when interpreting the permeation date, we applied the described resistance model by Henis and Tripodi ref 37.
More characterizations about the MOF template should be provided to better explain the formation of SURGEL membranes, such as the EDS of membrane cross-section (SURMOF and SURGEL).
A3. As stated above we added XRD and IRRAS characterization of the SURMOF, the crosslinked SURMOF as well as the SURGEL to the supporting information Figures S1 and S2. Together with the already included TOF-SIMS, AFM and SEM data in the manuscript we believe the material characterization is now well proven.
The references about hybrid membrane with microporous materials were not sufficiently cited and the authors can add some recent literatures like J. Membr. Sci., 570-571 (2019) 278-285;J. Mater. Chem. A, 7 (2019) 10898-10904 for enrich the content.
A4. We thank the reviewer for his suggestion to improve the citation of hybrid membranes with microporous materials and added the mentioned references (ref. 7 and ref. 8 in the revised version).
The coordinate axis of figures in this paper (especially fig.3 and fig.4) are very small and hard to recognize.
A5. Thank you for your comment, we enhanced the Figure quality and visibility in the revised manuscript.
There is something wrong with the typesetting of paper, and please check the whole manuscript again and make some necessary improvements.
A6. We corrected the typesetting in the revised manuscript.
Reviewer 2 Report
Thank you for the opportunity to review the interesting manuscript "Synthesis, transfer, and gas separation characteristics of MOF-templated polymer membranes". I believe that this experimental contribution to membrane preparation and separation property of MOF-templated membranes justifies the acceptance and publication of this manuscript in Membranes. However, I also believe that the manuscript should go through minor but important revisions delineated below:
1) In Introduction, the authors should add why the authors select PMMA, PDMS, PAN for membrane preparation. The authors should describe the gas permeance performances and challenges of the pristine polymer membranes and discuss the comparison between the pristine polymer membranes and the MOF-templated membarnes.
2) The measured XRD patterns should be provided to discuss the structure of MOF and microstructure of membranes with structure transferred by MOF.
3) The authors should measure N2 adsorption/desorption isotherms for SURMOF and crosslinking SURMOF to discuss the change of microstructure after the crosslinking step.
4) The measured chemical compositions should be provided to discuss whether there is residual metal. I wonder how the authors determine the removal of metal from the products.
5) The authors should measure TGA for discussion about the structural stability of MOF and MOF-templated membranes.
6) The authors should provide the measured gas permeances at each temperature. The discussion on the contribution of dissolution and diffusion for gas permeance should be added.
Following these improvements, I would strongly encourage the acceptance and publication of this work.
Author Response
We would like to thank the reviewers for their time to evaluate our manuscript for publication in Membranes. We highly appreciate all the reviewers’ valuable comments. In compliance with the reviewer’s comments and suggestions, we wish to address these points with additional experimental evidence where applicable and we have revised the manuscript accordingly.
Please find below a point-by-point answers and responses to the reviewer’s comments and suggestions; the reviewers’ comments are reproduced unchanged with our response included. We included a new supporting information file
Point-by-point response to reviewers:
REVIEWER 2
Comments and Suggestions for Authors
Thank you for the opportunity to review the interesting manuscript "Synthesis, transfer, and gas separation characteristics of MOF-templated polymer membranes". I believe that this experimental contribution to membrane preparation and separation property of MOF-templated membranes justifies the acceptance and publication of this manuscript in Membranes. However, I also believe that the manuscript should go through minor but important revisions delineated below:
1) In Introduction, the authors should add why the authors select PMMA, PDMS, PAN for membrane preparation. The authors should describe the gas permeance performances and challenges of the pristine polymer membranes and discuss the comparison between the pristine polymer membranes and the MOF-templated membarnes.
A1. PAN porous membrane was chosen as a support for the SURGEL membrane due to its high reproducibility, mechanical and chemical stability and high gas transport properties, determined by the surface porosity of 13-15% and small layer thickness about 40µm.
PDMS was selected as an intermediate layer between the selective and porous layers for the following considerations: a) the layer should be insoluble in organic solvents – PDMS layer is cross-linked after layer deposition; b) should provide as low as possible resistance for gas transport from the selective layer to the porous substrate – PDMS is one of the “fastest” polymers with relatively low selectivity for permanent gases. [I. Cabasso, K.A. Lundy, Method of Making Membranes for Gas Separation and the Composite Membranes US Patent 4,602,922 (1986)]. The deposition technique developed at HZG [Peter, J.; Peinemann, K.-V. V. Multilayer composite membranes for gas separation based on crosslinked PTMSP gutter layer and partially crosslinked Matrimid® 5218 selective layer. J. Memb. Sci. 2009, 340, 62–72, doi:http://dx.doi.org/10.1016/j.memsci.2009.05.009.] allows formation of 100-150nm thick defect free layer of highly cross-linked PDMS with improved adhesion properties in batches up to 160m2.
PMMA and PS were chosen as transfer layer for a few nm thick SURGEL layers since they are inert in terms of gas transport but sufficiently mechanically stable to serve.
The manuscript contains the discussion on the support membranes in section 3.2:
“For gas permeation experiments, we prepared thin film composite (TFC) membranes,35-36 comprised of PDMS/PAN and SURGEL as selective layer. We chose the PDMS/PAN support to enhance the mechanical stability to the nanometer thick SURGEL and minimize roughness-induced strain.30 The PDMS layer is also diminishing the effect of viscous gas flow through larger defects in the SURGEL layer. To evaluate the membrane properties of the individual SURGEL layer we utilized the resistance model introduced by Henis and Tripodi.37 In previous works, such an analysis has been successfully applied for the determination of properties of nanometer thin CNMs30 and CMPs.22“
2) The measured XRD patterns should be provided to discuss the structure of MOF and microstructure of membranes with structure transferred by MOF.
A2. We thank the reviewer for her/his suggestion and added the XRD pattern and discussion to the supporting information Figure S1.
3) The authors should measure N2 adsorption/desorption isotherms for SURMOF and crosslinking SURMOF to discuss the change of microstructure after the crosslinking step.
A3. Thank you for this interesting suggestion. However, for our films (of only about 50 nm thickness) the amount necessary to reliably measure the N2 sorption isotherms is not feasible. Only with specialized equipment such measurements are possible (e.g. a Quartz Crystal Microbalance (QCM) operated at low temperature) which are not available within reasonable time. However, we agree that such data would be interesting and would be worth investigating to further analyze the system and get a deeper understanding, which we anticipate in future studies.
4) The measured chemical compositions should be provided to discuss whether there is residual metal. I wonder how the authors determine the removal of metal from the products.
A4. We thank the reviewer for this question. We can follow the removal of the metal ions on the one hand via the added IR data in the supporting information Figure S2. The carboxylate bands of the SURMOF linker when coordinated to the metal are: −COO–, 1610–1550 cm–1 for asymmetric vibration and 1420–1300 cm–1 for symmetric vibration. After metal removal, these carboxylate bands shift to about 1710-1750 cm-1 which corresponds to the carbonyl groups contained in the protonated DA-SBDC linker. Therefore indicating that the carboxylate groups are no longer coordinated to the copper ions. Additionally, in EDX investigations of SURGEL films we did not detect any residual copper signal in similar samples as shown in reference 9 (J. Am. Chem. Soc. 2014, 136 (1), 8-11).
5) The authors should measure TGA for discussion about the structural stability of MOF and MOF-templated membranes.
A5. Thank you for this interesting suggestion. However, the amount necessary to reliably measure TGA (~10 mg) not feasible.
6) The authors should provide the measured gas permeances at each temperature. The discussion on the contribution of dissolution and diffusion for gas permeance should be added.
A.6 We included the measured gas permeances at each temperature in the supporting information Table S1.
However, due to the low thickness of the synthesized selective layer, it is technically not possible to obtain the data on diffusion and solubility coefficients during the gas transport measurements. For example: taking the selective layer thickness as 50nm (5E-6cm); permeability coefficient for the gas with lowest diffusion coefficient CH4 as e.g. 0.1 Barrer (1E-11 cm3(STP) cm cm-2 s-1 cmHg-1) and solubility coefficient of CH4 as 1E-3 cm3(STP) cm-3 cmHg-1 one will obtain the value of the diffusion coefficient as 1E-8 cm2 s-1. For this diffusion coefficient and layer thickness, the time-lag will be 4.17E-4 s. Our experimental equipment has time resolution of 2ms and systematic error of 35±7ms, which makes determination of the diffusion coefficient for such thin layers impossible. The systematic error depends on the time, which is necessary for the valve on the feed side to be opened after the command to do so is issued by the software and on the gas nature (viscosity), which needs to be delivered from the orifice of the valve to the feed surface of the membrane. To conclude: it will be necessary to conduct special experiments on quartz microbalance with oscillation frequency of 1 or better 10 MHz in order to assess sorption and diffusion parameters of the developed materials.
Following these improvements, I would strongly encourage the acceptance and publication of this work.
Overall, we thank the reviewers for insight and we have revised the manuscript to clarify all the points. We believe that these changes have greatly contributed to improving our manuscript, which we hope the reviewer finds satisfactory.
Round 2
Reviewer 2 Report
Authors have revised the manuscript satisfactory as the reviewer pointed out.I consider this manuscript meets the criteria for the publication.